# Plant Growth-Promoting Bacteria from Tropical Soils: In Vitro Assessment of Functional Traits

**DOI:** 10.3390/microorganisms13102321

**Published:** 2025-10-07

**Authors:** Juliana F. Nunes, Maura S. R. A. da Silva, Natally F. R. de Oliveira, Carolina R. de Souza, Fernanda S. Arcenio, Bruno A. T. de Lima, Irene S. Coelho, Everaldo Zonta

**Affiliations:** 1Department of Soils, Institute of Agronomy, Federal Rural University of Rio de Janeiro, BR 465 Km07, Seropedica Rio de Janeiro 23897-000, Brazil; jufnunes2@gmail.com (J.F.N.); maura@ufrrj.br (M.S.R.A.d.S.); rochanatally515@gmail.com (N.F.R.d.O.); eurufinocarol@gmail.com (C.R.d.S.); bbbruno04@gmail.com (B.A.T.d.L.); 2Department of Veterinary Microbiology and Immunology, Veterinary Institute, Federal Rural University of Rio de Janeiro, BR 465 Km07, Seropedica Rio de Janeiro 23897-000, Brazil; feseixasufrrj@gmail.com (F.S.A.); irenecoelho@ufrrj.br (I.S.C.)

**Keywords:** biocontrol, bioprospecting, agricultural sustainability

## Abstract

Plant growth-promoting bacteria (PGPBs) offer a sustainable alternative for enhancing crop productivity in low-fertility tropical soils. In this study, 30 bacterial isolates were screened in vitro for multiple PGP traits, including phosphate solubilization (from aluminum phosphate—AlPO_4_ and thermophosphate), potassium release from phonolite rock, siderophore production, indole-3-acetic acid (IAA) synthesis, ACC deaminase activity, and antagonism against *Fusarium* spp. Statistical analysis revealed significant differences (*p* < 0.05) among the isolates. The most efficient isolates demonstrated a solubilization capacity ranging from 24.0 to 45.2 mg L^−1^ for thermophosphate and 21.7 to 23.5 mg L^−1^ for potassium from phonolite. Among them, *Pseudomonas azotoformans* K22 showed the highest AlPO_4_ solubilization (16.6 mg L^−1^). IAA production among the isolates varied widely, from 1.34 to 9.65 µg mL^−1^. Furthermore, 17 isolates produced carboxylate-type siderophores, and only *Pseudomonas aeruginosa* SS183 exhibited ACC deaminase activity, coupled with strong antifungal activity (91% inhibition). A composite performance index identified *P. azotoformans* K22, *E. hormaechei* SS150, *S. sciuri* SS120, and *B. cereus* SS18 and SS17 as the most promising isolates. This study provides a valuable foundation for characterizing plant growth-promoting traits and identifies key candidates for future validation and the development of microbial consortia.

## 1. Introduction

The continued growth of the global population has driven agricultural production to historically high levels, primarily sustained by the intensive use of agrochemicals. Among these inputs, synthetic fertilizers and pesticides stand out as indispensable pillars of modern agriculture, without which global agricultural production can be reduced by approximately 40–50% [1]. Nitrogen, phosphate, and potassium fertilizers are among the most widely used agricultural inputs, with nitrogen fertilizers accounting for between 55% and 60% of global consumption, followed by phosphate fertilizers at approximately 22% to 25%, and potassium fertilizers, responsible for 18% to 20% [2]. Furthermore, pesticides, including herbicides, insecticides, and fungicides, are essential in reducing losses caused by pests, pathogens, and invasive plants, which can compromise up to 40% of global agricultural production [3].

The intensive use of synthetic fertilizers and pesticides has had significant environmental and economic impacts. The dependence of phosphate and potassium fertilizers on non-renewable natural resources threatens long-term food security. In addition to being depleted, phosphate rock reserves are highly concentrated in a few countries, posing a geopolitical risk. Similarly, potassium deposits are abundant in a few countries, reinforcing the vulnerability of global supply chains [4,5]. Furthermore, nutrient leaching contaminates aquifers and water bodies, leading to eutrophication and, consequently, creating dead zones in aquatic ecosystems [6,7]. Synthetic pesticides, when poorly managed, affect non-target species, including indispensable pollinators such as bees, and compromise soil health by reducing microbial biodiversity [8,9,10].

Therefore, there is an urgent need for sustainable alternatives that allow diversification of nutrient sources and an increase in their efficiency. Most studies on the microbial solubilization of insoluble phosphates have evaluated calcium phosphates, such as hydroxyapatite and tricalcium phosphate, which are predominant in temperate soils; however, the results obtained from this approach have limited applicability in tropical soils [11]. In these environments, low phosphorus availability is strongly related to fixation in iron and aluminum oxides, with aluminum phosphate (AlPO_4_) being one of the least soluble and bioavailable forms, which remain little explored in research [12]. Furthermore, alternative sources of phosphate fertilizers, such as thermophosphate, offer gradual phosphorus release and reduce losses through fixation and leaching [13,14]. Similarly, a potassic rock abundant in Brazil, phonolite, has emerged as a strategic alternative for reducing external dependence on potassium and enhancing local resources [15,16].

Assessing the ability of native microorganisms to solubilize these mineral sources is an essential step toward developing bioinoculants adapted to tropical soil and climatic conditions [17]. In this context, plant growth-promoting bacteria (PGPBs) have emerged as promising alternatives for promoting agricultural sustainability [18]. These beneficial microorganisms contribute significantly to nutrient cycling, including the mineralization of organic matter, phosphate solubilization, potassium availability, biological nitrogen fixation, and the availability of micronutrients, such as iron, through siderophore production [19,20,21]. Furthermore, they can modulate plant development through the production of phytohormones, especially auxins [19,22], and contribute to tolerance to abiotic stresses through the production of ACC-deaminase, an enzyme that degrades the ethylene precursor (ACC), thereby preventing excessive accumulation of this hormone associated with stress responses [23].

In addition to promoting plant growth, PGPBs play a crucial role in the biological control of pathogens by acting as natural suppressors through multiple mechanisms. These microorganisms can inhibit the development of phytopathogenic fungi, bacteria, and nematodes through the production of antimicrobial compounds (such as antibiotics and lytic enzymes), competition for niches and nutrients, and induction of systemic resistance (ISR) in plants [24]. This multifaceted antagonistic activity is particularly valuable against harmful soil-borne pathogens such as *Fusarium* species, which cause devastating wilts, rots, and blights in numerous economically important crops. For example, *Fusarium oxysporum f. sp. lycopersici* causes Fusarium wilt, a disease that can lead to substantial economic losses in tomato production globally [25]. Furthermore, *Fusarium* species contaminate grains and food products with hazardous mycotoxins, such as fumonisins and deoxynivalenol (DON). These toxins pose a serious risk to food safety and can cause various health issues in humans and livestock, including immunosuppression and esophageal cancer, highlighting the critical need for effective biocontrol strategies [25].

Considering the challenges posed by tropical soils, which are often characterized by high acidity, high levels of toxic aluminum, and low availability of phosphorus and potassium, bioprospecting native PGPBs has become a priority. These microorganisms, which are naturally adapted to such conditions, represent a solid basis for the development of multifunctional bioinoculants capable of optimizing the use of available mineral sources in the soil, modulating plant growth through phytohormone synthesis, increasing tolerance to abiotic stresses, and contributing to the biological control of pathogens [17]. These characteristics position PGPBs as strategic tools for promoting more sustainable agricultural systems, aligning with the recommendations of the Intergovernmental Panel on Climate Change (IPCC) for climate change mitigation and adaptation [26]. Therefore, investing in exploring tropical microbiota transcends the scope of scientific opportunity; it is consolidated as a pressing need to advance toward truly resilient and innovative tropical agriculture. However, translating this potential into practical applications requires a systematic screening of native microorganisms for key agronomic traits. Therefore, this study aimed to evaluate the potential of native bacteria in relation to attributes, such as nutrient solubilization, siderophore production, phytohormone synthesis, ACC deaminase synthesis, and antagonistic activity against phytopathogens, thereby generating useful information to support future research to develop bioinoculants for tropical systems.

## 2. Materials and Methods

### 2.1. Origin of Bacterial Isolates

This study used 30 bacterial strains deposited in the Laboratory of Bacteriology and Molecular Biology of the Federal Rural University of Rio de Janeiro. Of these, 26 were isolated from the rhizospheres of plants naturally established in saline soils in the state of Rio de Janeiro, Brazil [27]. These isolates were selected based on their ability to adapt to adverse environmental conditions, a characteristic considered relevant for developing microbial inoculants. Additionally, four bacterial strains of different origins were included to expand the taxonomic and functional diversity of the study set. One Bacillus pumilus isolate was obtained from the Guandu River in the metropolitan region of Rio de Janeiro. An isolate of Pseudomonas azotroformans was obtained from the Biofertilizer PotaBARVAR-2, produced by Green Biotech Incorporation (Balehonnur, India), and formulated with a focus on potassium solubilization. The Bacillus thiuringiensis isolate was obtained from the commercial product Vectobac (Kirchhoffs, Cape Town, South Africa). The isolate Bacillus altitudinis complex CCT 3116 was acquired from the Culture Collection of the André Tosello Tropical Research and Technology Foundation, as it is a well-studied strain recognized for its biological properties.

### 2.2. Bacterial Identification by MALDI-TOF Mass Spectrometry

After reactivation on Tryptone Soy Broth (TSB) agar at 30 °C for 24 h, bacterial cultures were transferred to a 96-well microplate (MSP, Bruker, Billerica, MA, USA). A lysis solution comprising 70% formic acid (Sigma-Aldrich^®^, St. Louis, MO, USA) was added to the bacterial pellet in a volume sufficient to fully cover it. Subsequently, 1 μL of matrix solution comprising alpha-cyano-4-hydroxycinnamic acid diluted in 50% acetonitrile and 2.5% trifluoroacetic acid (Sigma-Aldrich^®^) was applied to the bacterial extract. Mass spectra were acquired using a MALDI-TOF LT Microflex mass spectrometer (Bruker^®^, Billerica, MA, USA), operating in linear mode with a 337 nm nitrogen laser, and controlled by the FlexControl software version 3.3 (Bruker^®^). Spectra were collected in the mass range of 2000–20,000 m/s and subsequently analyzed using the MALDI Biotyper 2.0 program (Bruker^®^), with standardized settings for bacterial identification. The program compares unknown sample spectra with reference samples in a database.

Values greater than 2.0 indicate high identification reliability. In the range of 1.7 to 1.99, the identification reliability was low. Values below 1.7 indicate no identification reliability.

### 2.3. Inoculum Preparation

Bacterial isolates were grown on nutrient agar plates (HiMedia, Mumbai, India) and incubated at 30 °C for 24 h. A single colony of each strain was inoculated into test tubes containing 9 mL of TSB (Kasvi, Curitiba, Brazil). The cultures were incubated at 30 °C for 24 h on an orbital shaker at 150 rpm. Then, the absorbance of each culture was measured using a spectrophotometer at 600 nm and adjusted to values between 0.9 and 1.0 by dilution in sterile TSB.

### 2.4. Solubilization of Phosphate Sources

The isolates were prepared as described in Section 2.3. After this step, 300 µL of each bacterial culture was inoculated into 50 mL Falcon tubes containing 35 mL of NBRIP medium (National Botanic Research Institute’s phosphate) (10 g·L^−1^ glucose; 0.15 g·L^−1^ (NH_4_)2SO_4_; 0.2 g·L^−1^ KCl; 5 g·L^−1^ MgCl_2_·6H_2_O; 0.25 g·L^−1^ MgSO_4_·7H_2_O; phosphate source) supplemented with 5 g·L^−1^ of aluminum phosphate (AlPO_4_) or 13 g·L^−1^ of thermophosphate (Yoorin Fertilizers, São Paulo, Brazil), with an initial pH of 7.0, and maintained under stirring at 150 rpm for 7 d [28]. The tests were performed in triplicate.

A 10 mL aliquot of each sample unit was transferred to 15 mL Falcon tubes and centrifuged at 6000 × *g* for 10 min. The supernatant was filtered using a 0.22-µm membrane syringe filter to remove bacterial cells from the medium. The filtrate was used to determine the pH and diluted as needed to determine the soluble phosphorus. The soluble phosphorus concentration was assessed using the method described by Teixeira et al. [29]. Finally, the OD was measured by spectrophotometry at 660 nm. Control flasks containing uninoculated culture medium were used. Values for inoculated treatments were corrected by subtracting the values of their respective uninoculated controls. The standard curve was generated as described by Teixeira et al. [29]. The *Gluconacetobacter diazotrophicus* (BR-11281) strain was used as a positive control [30]. This strain was kindly provided by EMBRAPA Agrobiologia (Seropedica, Brazil).

### 2.5. Potassium Solubilization

The isolates were prepared as described in Section 2.3. After this step, 300 µL of each culture was transferred to 15 mL Falcon tubes containing 10 mL of modified Aleksandrov medium (5 g·L^−1^ glucose; 0.5 g·L^−1^ MgSO_4_.7H_2_O; 0.005 g·L^−1^ FeCl_3_; 0.1 g·L^−1^ CaCO_3_; 2.72 g·L^−1^ Na_2_ HPO_4_), supplemented with 2 g·L^−1^ potassium aluminum silicate (phonolite), with an initial pH of 7.0 [31]. The tubes were incubated at 30 °C and 150 rpm for 7 d. The tests were performed in triplicate.

Subsequently, the samples were centrifuged at 6000× *g* for 10 min. The supernatant was used to determine the pH and the amount of soluble potassium released into the solution. Flame photometry was used to determine the potassium content of the samples. Control flasks containing uninoculated culture medium were used. Values obtained from the uninoculated controls were subtracted from those of the respective inoculated treatments. The standard curve was prepared with the 1000 mg·L^−1^ K + standard solution with distilled water, as Silva [32] described.

The phonolite, whose main chemical composition is characterized by the major oxides 52.7% SiO_2_, 20.0% Al_2_O_3_, 8.1% K_2_O, 7.9% Na_2_O, 4.0% Fe_2_O_3_, 1.3% CaO, and 0.34% MgO, was subjected to comminution in a jaw crusher. This process aimed to reduce the particle size and increase the surface area of the material according to the methodology described by Dias [33].

### 2.6. Siderophore Production

Siderophore production was evaluated as described by Schwyn and Neilands [34]. The inocula were prepared as described in Section 2.3. After this step, 7 µL of each culture was inoculated into Petri dishes containing solid King B medium (3.0 g·L^−1^ glycerol; 4.0 g·L^−1^ bacteriological peptone; 0.23 g·L^−1^ K_2_HPO_4_; 0.30 g·L^−1^ MgSO_4_·7H_2_O; 15.0 g·L^−1^ Noble agar), adjusted to pH 6.8. The plates were incubated at 30 °C for 24 h. The chromium reagent Azurol S (CAS) was prepared as described below using a combination of three solutions:

Solution I: 60.5 mg of Chrome Azurol S (CAS) dissolved in 50 mL of distilled water.

Solution II: 27 mg of FeCl_3_·6H_2_O dissolved in 10 mL of 10 mM HCl.

Solution III: 72.9 mg of hexadecyltrimethylammonium bromide (CTAB) was dissolved in 40 mL of distilled water.

Initially, Solution I was mixed with 9 mL of Solution II. This mixture was then combined with Solution III, resulting in a blue solution. The CAS solution was then autoclaved at 110 °C for 10 min. The sterile CAS solution was added to King B culture medium at approximately 55 °C to prepare the assay medium in a 1:10 (*v*/*v*) ratio.

After incubation, siderophore production was determined by observing the characteristic color halos around the colonies which were classified as follows: catechol, transition from blue to pink; hydroxamate, transition from blue to orange; and carboxylate, transition from blue to light yellow [35].

### 2.7. Production of Indole-3-Acetic Acid (IAA)

Salkowski colorimetric assay was used to quantitatively analyze IAA [36]. Bacterial isolates were prepared as described in Section 2.3. Subsequently, a 50 μL aliquot of the culture was inoculated in TSB medium supplemented with 1 g·L^−1^ of L-tryptophan (Sigma-Aldrich, St. Louis, MO, USA), and the tubes were incubated for 72 h at 30 °C at 150 rpm. The test was performed in triplicate. Samples (2 mL) were collected, and the cells were centrifuged (13,000 × *g*, 5 min). One milliliter of the resulting supernatants was mixed with 2 mL of Salkowski reagent (1 mL of 0.5 M FeCl_3_ in 50 mL of 35% (*v*/*v*) HClO_4_) and incubated at 28 °C in the dark for 30 min. The absorbance was measured using a spectrophotometer at 540 nm. A standard curve was constructed as described by Lana et al. [37]. *Azospirillum brasilense* strain (BR-11001) kindly provided by EMBRAPA Agrobiologia, was used as the positive control [38].

### 2.8. Production of ACC-Deaminase

Bacterial isolates were prepared as described in Section 2.3. The isolates were then centrifuged at 3000× *g* for 5 min. The pellet was washed twice with a 0.1 M solution. The pellet was resuspended in 1 mL of 0.1 M Tris-HCl and 2 μL was inoculated in DF medium (2.0 g·L^−1^ glucose; 2.0 g·L^−1^ citric acid; 4.0 g·L^−1^ KH_2_·PO_4_; 6.0 g·L^−1^ Na_2_·HPO_4_; 0.2 g·L^−1^ MgSO_4_·7H_2_O; 20 g·L^−1^ agar) pH 7.2 [39].

DF medium with a nitrogen source (supplemented with 2 g·L^−1^ (NH_4_)_2_SO_4_) was used as a negative control, and DF medium without a nitrogen source was used. To evaluate the ACC-deaminase production capacity, DF medium without a nitrogen source was used, applying 120 μL of a 0.5-M ACC-deaminase solution to the surface of the plate. The plates were incubated at 30 °C for 7 d. Isolates that grew well in medium without nitrogen and with ACC deaminase solution were considered positive. All experiments were performed in triplicate. The *Herbaspirillum seropedicae* strain (BR-11790) provided by EMBRAPA Agrobiologia was used as the positive control [40].

### 2.9. Antagonist Activity

The bacterial isolates, prepared as described in Section 2.3, were evaluated for antagonistic activity against the phytopathogenic fungus *Fusarium oxysporum* f. sp. *lycopersici* (L3304), obtained from the Seed Epidemiology Laboratory of the Federal Rural University of Rio de Janeiro. The *Fusarium* sp. isolate was cultured on Potato Dextrose Agar (PDA) medium (HiMedia, Mumbai, India) for 7 d at 28 °C. PDA was selected for its optimal support of fungal mycelial growth, ensuring robust fungal biomass for antagonism assays.

Antagonism was determined using the dual-culture containment technique following the methodology of Fernandes et al. [41]. The bacterial strains were plated on PDA, forming four parallel strips spaced 40 mm apart. A 7 mm diameter disk of the culture medium containing the fungal mycelium was then placed in the center of the plate, perpendicular to and equidistant from the bacterial strips.

The plates were incubated at room temperature for 15 d. The experimental design included control plates consisting of fungal disks incubated in the absence of bacteria. Antagonistic activity was interpreted by analyzing colony area and percentage inhibition. The fungal colony area was obtained by measuring radial growth along the orthogonal axes. The mean of these two measurements was calculated and assumed as the radius (r) for area calculation using the formula πr^2^. The results are expressed in mm^2^. The percentage inhibition was determined as the ratio of the pathogen growth area in the presence of each bacterium to the area occupied by the pathogen in the control treatment. Using these values, the percentage inhibition (*% In*) was calculated using the following formula adapted from Fernandes et al. [41]:% In = 100 − [(treatment area mm) × 100)/(control area mm)

Based on the percentage inhibition values, bacterial isolates were classified into three levels of antagonism: low (<40%), moderate (40–80%), and high (>80%).

### 2.10. Statistical Analysis

The experiment was conducted using a completely randomized design. All statistical analyses were performed using R Core Team (2024) software. The normality of residuals was verified using the Shapiro–Wilk test, and the homoscedasticity of variances was verified using the Bartlett test. Data transformation was performed through automatic selection using the BestNormalize function in R software (version 4.4.2; R Core Team, 2024) when necessary. The transformed data were subjected to assumption verification. Subsequently, analysis of variance was applied, and when significant by the F test (*p* < 0.05), multiple comparisons of means were performed using the Scott-Knott cluster test at 5% probability.

The mean value of each variable was calculated per isolate and standardized using the Z-score method (mean = 0, standard deviation = 1). A composite index was constructed from this standardization by averaging the Z-scores and used to rank the isolates. For multidimensional comparative analysis, the mean data for each variable were normalized to a 0–1 (min–max) scale and used to construct the radar plots. Five isolates with the highest composite indices were selected, and the radar plots represented their profiles.

## 3. Results

Of the 30 bacterial isolates, 25 (83.3%) were identified with high reliability by the MALDI-TOF technique, presenting scores higher than 2.00 (Table 1). Three isolates (Bp2, K22, and SS120) were identified with intermediate reliability scores between 1.70 and 1.99. One isolate (SS11) did not reach a reliable level of identification, with a score lower than 1.70. The isolates from reference culture collections CCT 3116 (*Bacillus altitudini*), BR11281 (*Gluconacetobacter diazotrophicus*), BR11001 (*Azospirillum brasilense*), and BR11790 (*Herbaspirillum seropedicae*) were not subjected to re-identification by MALDI-TOF, as their taxonomic identities had been previously validated and maintained under strict quality control by the institutions of origin.

Regarding the solubilization of phosphorus from aluminum phosphate (AlPO_4_), the bacterial isolates showed statistically significant differences. The *Pseudomonas azotoformans* isolate (K22) demonstrated remarkable performance, reaching 16.5 mg·L^−1^. Despite this high value, its performance was statistically inferior to that of the positive control, *Gluconacetobacter diazotrophicus* (BR11281), which solubilized 50.6 mg·L^−1^. In the thermophosphate evaluation, significant differences in solubilization capacity were observed between the isolates. *Bacillus isolates cereus* (SS18, SS17, SS80, SS68, SS36, SS101, SS29, SS89), *Bacillus altitudinis* (CCT 3116), and *Pseudomonas azotoformans* (K22) stood out with the highest solubilization values, ranging from 24.0 to 45.2 mg·L^−1^. All isolates performed better than the positive control, which solubilized 19.9 mg·L^−1^. Regarding potassium availability from phonolites, the highest solubilization was observed in the *Enterobacter* sp. SS11 (23.5 mg·L^−1^) and SS186 (23.5 mg·L^−1^), followed by *Pseudomonas aeruginosa* sp. SS183, *Pantoea* sp. SS150, *Enterobacter hormaechei* SS15, and *Staphylococcus sciuri* SS120, ranging from 22.8 mg·L^−1^ to 21.7 mg·L^−1^ (Figure 1).

All isolates promoted marked acidification of the medium with AlPO_4_ after seven days of incubation, with a final pH ranging from 3.5 to 5.2. Culture media inoculated with nine isolates had a final pH between 3.5 and 3.9, including SS11 and SS89 (both at pH 3.5). Culture media inoculated with the remaining 19 isolates had a final pH of 4.0 and 5.0. Despite this acidification, the correlation between pH and solubilization was very weak and not significant (R = −0.173; *p* = 0.097), suggesting that other mechanisms may be more relevant in the AlPO_4_ solubilization process. In the medium supplemented with thermophosphate, the final pH ranged from 5.0 to 8.6. Most isolates (13/30) acidified the medium, presenting a pH between 5.0 and 6.0, while 12 maintained the pH range between 6.0 and 7.0. Only three isolates remained close to the initial pH of the culture medium (7.0), and two exhibited results higher than the initial pH, including *Enterobacter* sp. (SS11, pH 8.6) and *Bacillus cereus* (SS89, pH 8.3). The correlation between pH and phosphorus solubilization from thermophosphate was moderately negative and statistically significant (R = −0.541; *p* < 0.000001), indicating that acidification favored the release of phosphorus from this source. In the media containing phonolite, the final pH ranged from 4.2 to 6.4, with the media inoculated with most isolates (19/30) maintaining a pH close to 5.0, eight resulting in a pH drop to near 4.0, and three maintaining a pH close to the initial value, such as *Bacillus cereus* (SS101, pH 6.4) and *Enterobacter hormaechei* (SS145, pH 6.3). The correlation between acidification and solubilization was moderately positive and highly significant (R = 0.391; *p* = 0.0003), confirming that pH reduction favors K release from phonolites (Figure 2).

Siderophore production: 17 isolates produced carboxylate-type siderophores, whereas the others did not produce this molecule or any other type of siderophore (Table 2). Regarding ACC deaminase activity, only *Pseudomonas aeruginosa* (SS183) tested positive, whereas all other isolates tested negative (Appendix A).

*Pseudomonas azotoformans* K22 and *Staphylococcus sciuri* SS120 stood out in IAA production (Figure 3), with 8.94 µg·mL^−1^ and 9.65 µg·mL^−1^, respectively. Still, these values were statistically lower than those of the positive control *Azospirillum brasilense* (BR11001), which produced 29.4 µg·mL^−1^, indicating that the reference strain has greater metabolic efficiency in synthesizing the phytohormone. In contrast, *Bacillus cereus* SS88, SS26, and SS31 showed very low production (1.56 µg·mL^−1^ to 1.43 µg·mL^−1^), suggesting limited capabilities in this plant growth promotion mechanism.

Regarding antagonistic activity against the phytopathogenic fungus *Fusarium* spp., the isolate *Pseudomonas aeruginosa* (SS183) showed the greatest inhibitory effect, with 91% inhibition of mycelial growth (Figure 3). Two isolates of *Bacillus cereus* (SS101 and SS137) exhibited moderate activity with 44% and 52%, respectively. The remaining isolates showed low inhibition percentages, less than 20%, with nine isolates showing no inhibitory effect.

Analysis of the 30 bacterial isolates based on the composite performance index (average standardized Z-scores) revealed differences in the functional potential of the evaluated strains (Figure 4). This index integrated key variables related to plant growth promotion and biocontrol, including antagonism, indole acetic acid (IAA) production, aluminum phosphate solubilization (P_AL), thermophosphate solubilization (P_T), and potassium solubilization (K). The ranking obtained allowed us to identify isolates with superior performance, highlighting the isolates of *P. azotoformans* K22, *E. hormaechei* SS150, and *S. sciuri* SS120, and two isolates of *B. cereus* SS18 and SS17, which presented the highest positive index values, suggesting that they are promising candidates for future biotechnological applications. Notably, owing to their qualitative nature, siderophore production and ACC deaminase attributes were not included in previous quantitative statistical analyses. However, we observed that, among the five isolates with superior performance, *P. azotoformans* K22 and two *B. cereus* isolates, SS18 and SS17, showed siderophore (carboxylate) production.

The five isolates that presented the best results in the evaluated growth-promoting attributes were compared using a radar chart highlighting their specific functional characteristics (Figure 5). The *P. azotoformans* K22 isolate presented the highest composite index, emphasizing IAA production and phosphorus solubilization in both sources analyzed. The *E. hormaechei* SS150 and S. sciuri SS120 isolates showed balanced performance across all metrics, with consistent values across all variables analyzed. The isolate *B. cereus* SS18 excelled in K availability, whereas *B. cereus* SS17 had an intermediate profile. The shape of the polygons in the radar plot indicated that these isolates had significant peaks, but with a notable contribution to antagonism. The polygon configuration in the graph highlights that these isolates have complementary functional potential, reinforcing their viability for use in bacterial consortium formulations capable of integrating multiple mechanisms for plant growth promotion and biocontrol.

## 4. Discussion

Bacterial identification using matrix-assisted laser desorption/ionization time-of-flight (MALDI-TOF) mass spectrometry has become a fast, reliable, and low-cost method for routine microbiology laboratories with revolutionary application in microbial taxonomy [42]. The high scores obtained in this study, especially for isolates of the *Bacillus cereus* complex, corroborate the robustness of this technique for discriminating between morphologically and genetically complex groups [43].

The functional attribute analyses of the studied bacteria revealed remarkable diversity in the metabolic capacity of the isolates, reinforcing that efficiency is a strain-dependent characteristic. In the solubilization of P from AlPO_4_, the *P. azotoformans* isolate K22 showed significant solubilization of P from AlPO_4_. This finding is strategic in tropical soils, characterized by high Al^3+^ contents and low P availability, where the solubilization of compounds such as AlPO_4_ is essential for agricultural sustainability. The absence of a linear correlation between acidification and solubilization suggests the participation of other complementary mechanisms in phosphorus solubilization in addition to the production of organic acids, such as the production of exopolysaccharides (EPS), which act synergistically to facilitate the release of adsorbed phosphate [11,19].

In the solubilization of P from thermophosphate, the results demonstrate that several bacteria, particularly strains of *B. cereus* (SS17, SS18, SS68), *P. azotoformans* (K22), and *Enterobacter* sp. (SS11), exhibited high solubilization capacity, releasing significant amounts of phosphorus (P) while modulating the pH of the medium. This process enhances the agronomic advantages of thermophosphate as a slow-release fertilizer, complementing its gradual dissolution and reducing losses because of fixation in highly weathered tropical soils [44]. Thus, the interaction between alternative fertilizers and solubilizing microorganisms can increase phosphorus use efficiency, reduce applied doses, and contribute to more sustainable agricultural practices.

Regarding potassium release from phonolites, the isolates of *E. hormaechei* (SS186) and *P. aeruginosa* (SS183) showed high solubilization capacities, which were positively correlated with environmental acidification. Phonolite has been studied as a strategic option to reduce dependence on conventional soluble fertilizers, such as potassium chloride (KCl). Research has shown that phonolites can gradually release their nutrients when subjected to fine grinding or microbial activation processes [45,46,47].

The production of IAA by some investigated bacterial strains, particularly *S. sciuri* (SS120) and *P. azotoformans* (K22), was comparable to that reported for commercial strains of *Azospirillum brasilense*, as documented by Fukami et al. [48], demonstrating remarkable potential for agricultural applications. IAA plays crucial roles in plant development by regulating cell elongation and lateral and adventitious root formation and mediating responses to environmental stresses. However, excessive IAA concentrations can have phytotoxic effects, highlighting the need to evaluate physiologically relevant doses in plant experiments [49]. Recent studies have explored potential applications of IAA-producing bacteria in agriculture [50]. Alori and Babalola (2017) reviewed the use of microbial inoculants to improve crop quality and health in Africa and highlighted the importance of microbial auxins [51]. Fukami et al. [48,49] demonstrate that the benefits of *Azospirillum* go far beyond biological nitrogen fixation, including significant contributions through the production of phytohormones.

The detection of the ACC deaminase enzyme only in *Pseudomonas aeruginosa* (SS183) deserves attention because this mechanism is recognized as central to mitigating biotic and abiotic stresses in plants. This enzyme degrades 1-aminocyclopropane-1-carboxylic acid (ACC), the immediate precursor of ethylene, and negatively regulates its synthesis under stress conditions. Bacterial ACC deaminase activity reduces ACC levels, decreases ethylene production, and allows plants to grow even under adverse conditions [19,52].

Producing carboxylate-type siderophores by 17 isolates reflects a fundamental adaptive strategy in tropical soils, where low ferric iron (Fe^3+^) availability is a limited factor. Rhizospheric bacteria use these high-affinity chelators to solubilize and sequester iron, making it bioavailable. Notably, the genus *Pseudomonas* is a ubiquitous and efficient producer of pyoverdine-type siderophores, mixed-structure carboxylates whose biosynthesis is tightly regulated by iron deficiency and confers a crucial competitive advantage in oligotrophic environments, as detailed in Schalk et al.’s review [53]. This mechanism of “war by iron” (iron warfare) is a potent means of biocontrol, where deprivation of this essential nutrient directly suppresses the growth of fungal pathogens [53,54].

In antagonism toward *Fusarium* sp., the high inhibition rates observed for *P. aeruginosa* SS183 and, to a lesser extent, for *Bacillus cereus* SS137 confirm the potential of these strains as biocontrol agents. While *Pseudomonas* is recognized for the production of antifungal metabolites such as phenazines and pyocyanin [55], *Bacillus* species stand out for their arsenal of lipopeptides and hydrolytic enzymes with proven action against filamentous fungi [56,57]

Fusarium species are responsible for many diseases, such as vascular wilt and root rot, which result in significant global agricultural losses of essential crops. Furthermore, their ability to produce mycotoxins poses a serious risk to human and animal health [25]. Historical reliance on chemical control has proven increasingly unsustainable. The emergence of *Fusarium* spp. populations resistant to important fungicide classes such as triazoles is a widespread and documented problem that compromises control effectiveness [58]. This resistance emerges through an evolutionary process, driven by intense selective pressure from the repeated application of fungicides. This issue is further exacerbated by practices such as sublethal dosing and a lack of rotation among fungicide classes, ultimately leading to widespread control failures, rising production costs, and increased environmental damage [59].

Additionally, growing environmental and food safety concerns related to pesticide residues drive the global demand for safer and more sustainable alternatives within sustainable agriculture. Biological control is a fundamental pillar of Integrated Disease Management (IDM) [60]. Our results position the isolate *Pseudomonas aeruginosa* SS183 as a promising biocontrol candidate. The inhibition rate of 91% was remarkable and was in line with the best results reported for antagonistic strains of this genus [61].

In the present study, integrated statistical analysis using a composite index (average Z-score) identified five isolates with superior overall performance, each exhibiting distinct and complementary combinations of plant growth-promoting attributes. *Pseudomonas azotoformans* K22 demonstrated significant solubilization capacity for multiple phosphate and potassium sources (36.5 mg·L^−1^ thermophosphate, 16.6 mg·L^−1^ aluminum phosphate, and 16.7 mg·L^−1^ phonolite), associated with the production of IAA (8.94 mg·L^−1^) and carboxylate-type siderophore. However, it did not exhibit antagonist or ACC-deaminase activity. Similarly, *Bacillus cereus* SS17 stood out in the solubilization of thermophosphate (42.5 mg·L^−1^) and phonolite (14.2 mg·L^−1^), in addition to producing IAA (6.79 mg·L^−1^) and siderophore, but with low antagonist activity (5%) and absence of ACC-deaminase. The isolate B. cereus SS18 was the most efficient in the solubilization of thermophosphate (45.2 mg·L^−1^), although with moderate performance in the other sources and production of IAA (6.76 mg·L^−1^), but also lacking other attributes. *Staphylococcus sciuri* SS120 stood out as the largest producer of IAA (9.65 mg·L^−1^) and in the solubilization of phonolite (21.7 mg·L^−1^). In comparison, *Enterobacter hormaechei* SS150 exhibited the highest efficiency for phonolite (22.4 mg·L^−1^), although it had low performance on aluminum phosphate (0.8 mg·L^−1^), and intermediate IAA production (7.68 mg·L^−1^). Neither strain produced ACC-deaminase or siderophores and had low antagonist activity (15% and 8%, respectively). This analysis demonstrated that although no isolate individually presented excellence in all evaluated attributes, selection based on the composite index identified those with the most balanced and complementary combination of characteristics. These isolates compensated for specific deficiencies with high efficiency in key mechanisms of plant growth promotion. This approach revealed that integrated performance constitutes the most appropriate criterion for selecting microorganisms with potential applications as inoculants, as it reflects the multifactorial nature inherent in plant growth-promotion processes in real environments.

Considering the complementary nature of these attributes among isolates, strategies such as forming bacterial consortia and synergistic combinations between different genera or species have emerged as promising alternatives to overcome individual limitations [62,63,64,65]. Previous studies have demonstrated that consortia involving *Azospirillum brasilense* and *Pseudomonas fluorescens* promote synergistic positive effects on maize growth [64], just as microbial production of IAA can enhance other growth-promoting mechanisms [65].

This diverse functional profile represents strategic potential for developing bioinoculants adapted to the edaphoclimatic conditions of tropical soils. This perspective aligns with the targets of Sustainable Development Goal (SDG) 2 (Zero Hunger), which promotes more productive and nutrient-efficient agricultural systems, and SDG 13 (Climate Action), which contributes to reducing dependence on chemical fertilizers with a high environmental footprint [66,67].

Despite advances in the study of plant growth-promoting bacteria, it is important to recognize that in vitro assays often fail to capture the complexity of rhizosphere interactions under natural conditions. Factors such as microbial competition, soil variability, and the influence of the native microbiome can limit the efficacy observed in the laboratory [68]. Therefore, translating these promising findings into practical applications requires additional studies integrating agronomic efficiency, sustainability, and ecosystem resilience to develop robust and environmentally friendly microbial solutions.

## 5. Conclusions

This study shows that the analyzed bacteria produce plant growth attributes ranging from nutrient solubilization to hormonal modulation and biological control of pathogens. Of particular note are *Pseudomonas azotoformans* (K22), *Enterobacter hormaechei* SS150, *Staphylococcus sciuri* SS120, and two *Bacillus cereus* isolates (SS18 and SS17), which demonstrated the most balanced and complementary profiles of attributes in relation to the characteristics tested. Additionally, the SS183 strain of *Pseudomonas aeruginosa*, owing to its production of the ACC deaminase enzyme, an exclusive attribute in this study, and its high antagonistic capacity against the phytopathogen *Fusarium* sp., exhibit characteristics that, although not present in the isolates with the best overall performance, represent valuable functionalities for the composition of synergistic microbial consortia.

Future work should explore the synergy between these lineages in planta experiments and investigate the genes underlying their functional capabilities, paving the way for the bioengineering of even more efficient microorganisms.

## Figures and Tables

**Figure 1 microorganisms-13-02321-f001:**
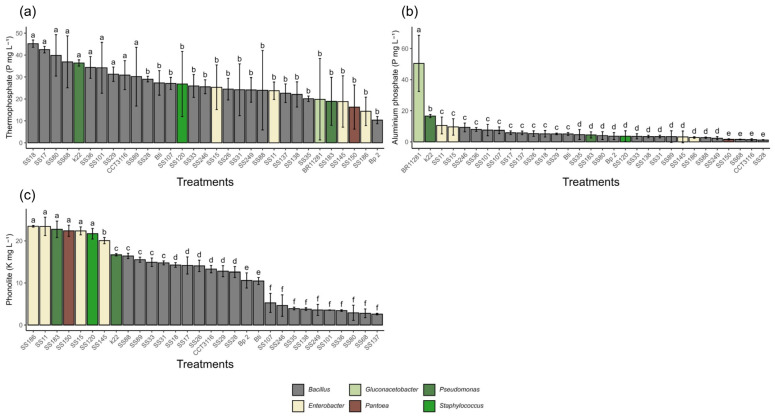
Solubilization of different mineral sources by different bacterial isolates after 7 days of incubation. (**a**) Phosphorus solubilization from thermophosphate. (**b**) Phosphorus solubilization from aluminum phosphate (AlPO_4_). (**c**) Potassium solubilization from phonolite. All values are expressed in mg L^−1^. Means followed by the same letter do not differ statistically from each other according to the Scott-Knott test at 5% probability.

**Figure 2 microorganisms-13-02321-f002:**
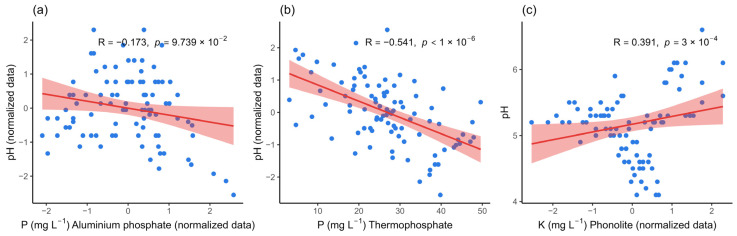
Correlation between pH values and solubilization of mineral sources by bacterial isolates after 7 days of incubation. (**a**) Correlation between culture medium pH and the solubilization of aluminum phosphate, (**b**) thermophosphate, and (**c**) potassium. The blue points represent the observations. The red line indicates the linear trend, and the pink band represents the 95% confidence interval for that trend. R is the Pearson correlation coefficient, and p is the *p*-value for the two-tailed test. *p*-value less than 0.05 indicate statistically significant correlations.

**Figure 3 microorganisms-13-02321-f003:**
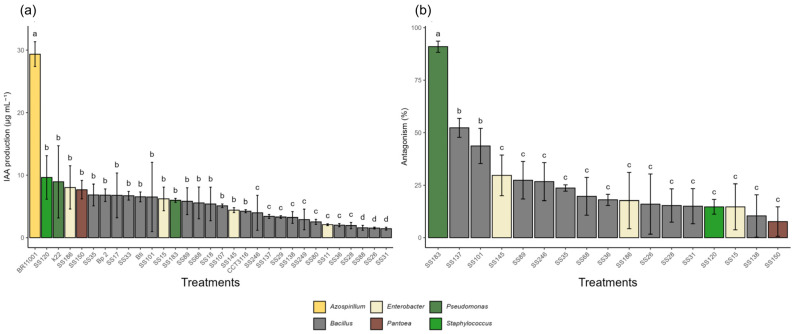
Indoleacetic acid (IAA) production and antagonistic activity by bacterial isolates. (**a**) IAA production (µg·mL^−1^) by bacterial isolates grown in tryptophan-supplemented medium. (**b**) Antagonistic activity (% inhibition) against *Fusarium* spp. Isolates that did not show antagonistic activity (% inhibition = 0%) are not represented in the graph. Different letters above the bars indicate statistical differences by the Scott-Knott test at 5% probability.

**Figure 4 microorganisms-13-02321-f004:**
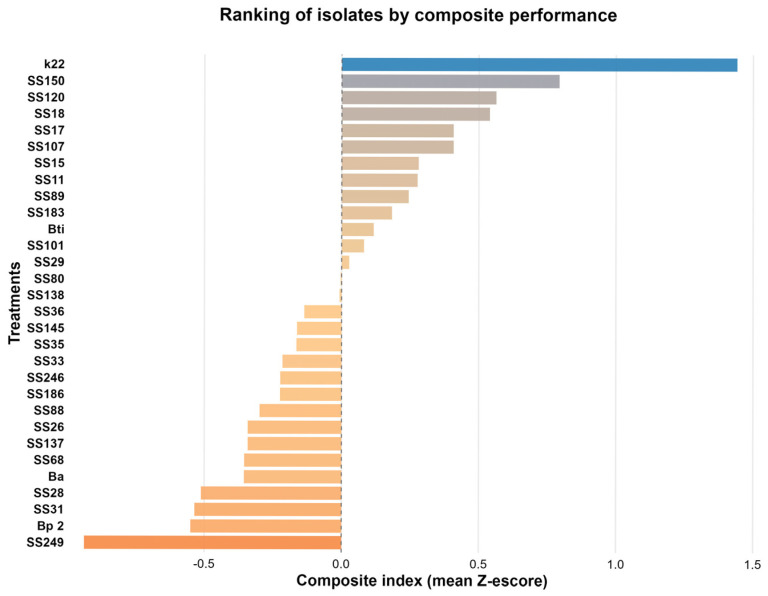
Ranking of bacterial isolates based on the composite performance index. The composite index was calculated by standardizing (Z-score method) the mean values of all analyzed variables (P solubilization from aluminum phosphate, P solubilization from thermophosphate, potassium solubilization from phonolite, indoleacetic acid production, and antagonist activity). The final index corresponds to the mean Z-score of each isolate, allowing the ordering of treatments according to their multifunctional performance. Isolates are organized from highest (top) to lowest (bottom) composite performance. The colors represent the relative values of the composite performance index. A divergent color scale was applied (orange–beige–blue), where warmer colors (orange) indicate isolates with lower index values, beige indicates values close to the median, and blue indicates isolates with higher index values.

**Figure 5 microorganisms-13-02321-f005:**
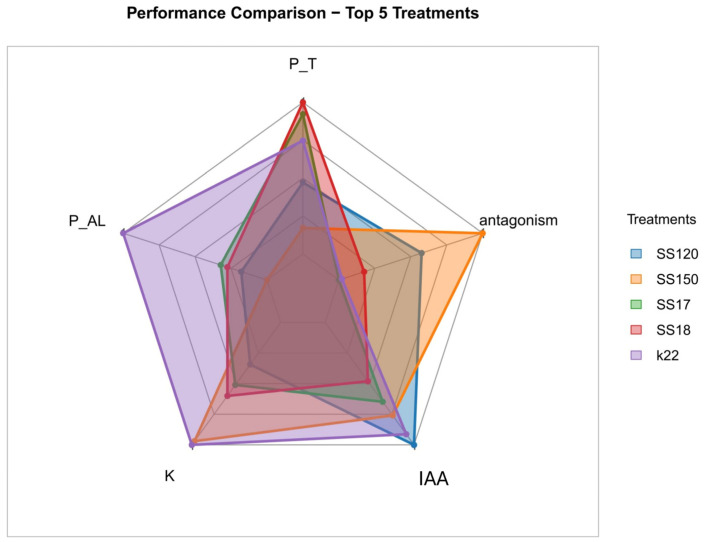
Multivariate comparative analysis of the five bacterial isolates with the highest composite performance indices. The variables represented are P solubilization from aluminum phosphate (P_AL), P solubilization from thermophosphate (P_T), K solubilization from phonolite (K), indole-3-acetic acid (IAA) production, and antagonist activity.

**Table 1 microorganisms-13-02321-t001:** Bacterial species identified by MALDI-TOF MS and their respective identification scores.

Code	Identification	Score	Code	Identification	Score
SS11	*Enterobacter* sp.	1.83	SS101	*Bacillus cereus*	2.44
SS15	*Enterobacter hormaechei*	2.00	SS107	*Bacillus cereus*	2.30
SS17	*Bacillus cereus*	1.82	SS120	*Staphylococcus sciuri*	2.09
SS18	*Bacillus cereus*	1.66	SS137	*Bacillus cereus*	2.27
SS26	*Bacillus cereus*	2.20	SS138	*Bacillus cereus*	2.05
SS28	*Bacillus cereus*	2.15	SS145	*Enterobacter hormaechei*	1.99
SS29	*Bacillus cereus*	2.19	SS150	*Pantoea* sp.	2.32
SS31	*Bacillus cereus*	2.27	SS183	*Pseudomonas aeruginosa*	2.37
SS33	*Bacillus cereus*	2.41	SS186	*Enterobacter hormaechei*	2.44
SS35	*Bacillus cereus*	2.13	SS246	*Bacillus cereus*	2.41
SS36	*Bacillus cereus*	2.31	SS249	*Bacillus cereus*	2.45
SS68	*Bacillus cereus*	2.29	Bp2	*Bacillus pumillus*	2.38
SS80	*Bacillus cereus*	2.31	Bti	*Bacillus thunrigiensis*	2.14
SS88	*Bacillus cereus*	2.49	K22	*Pseudomonas azotoformans*	2.36
SS89	*Bacillus cereus*	2.13			

**Table 2 microorganisms-13-02321-t002:** Siderophore production by bacterial isolates.

Code	Identification	Siderophore	Code	Identification	Siderophore
SS11	*Enterobacter* sp.	-	SS101	*Bacillus cereus*	-
SS15	*Enterobacter hormaechei*	Carboxylate	SS107	*Bacillus cereus*	Carboxylate
SS17	*Bacillus cereus*	Carboxylate	SS120	*Staphylococcus sciuri*	-
SS18	*Bacillus cereus*	Carboxylate	SS137	*Bacillus cereus*	Carboxylate
SS26	*Bacillus cereus*	-	SS138	*Bacillus cereus*	Carboxylate
SS28	*Bacillus cereus*	Carboxylate	SS145	*Enterobacter hormaechei*	Carboxylate
SS29	*Bacillus cereus*	Carboxylate	SS150	*Pantoea* sp.	-
SS31	*Bacillus cereus*	-	SS183	*Pseudomonas aeruginosa*	Carboxylate
SS33	*Bacillus cereus*	-	SS186	*Enterobacter hormaechei*	Carboxylate
SS35	*Bacillus cereus*	-	SS246	*Bacillus cereus*	-
SS36	*Bacillus cereus*	Carboxylate	SS249	*Bacillus cereus*	Carboxylate
SS68	*Bacillus cereus*	-	CCT 3116	*Bacillus altitudinis*	Carboxylate
SS80	*Bacillus cereus*	-	Bp2	*Bacillus pumillus*	Carboxylate
SS88	*Bacillus cereus*	-	Bti	*Bacillus thunrigiensis*	Carboxylate
SS89	*Bacillus cereus*	-	K22	*Pseudomonas azotoformans*	Carboxylate

The symbol (-) indicates no detectable production under the experimental conditions.

## Data Availability

The original data presented in the study are openly available in Zenodo at https://doi.org/10.5281/zenodo.17020394.

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
