# Peer review of "Plant Growth-Promoting Bacteria from Tropical Soils: In Vitro Assessment of Functional Traits"

_microorganisms, 2025, doi:10.3390/microorganisms13102321_

Round 1
Reviewer 1 Report
Comments and Suggestions for Authors
There are too many references. Consider to elimimate those redundant
This manuscript explores the tropical agriculture soils looking for plant growth-promoting bacteria (PGPB) and antagonist against phytopathogen Fusarium spp. The aim was to select isolates showing these attributes (PGPB and Fusarium antagonism) to use them as biofertilizer and biological control of Fusarium, to this system.
Authors performed in vitro assays to evaluate the ability of 30 isolates for nutrient solubilization, siderophore, and phytohormone (indole acetic acid) production, ACC deaminase synthesis, and antagonistic activity against the phytopathogen fungus Fusarium.
Authors claim that, out of the isolates tested, P. azotoformans (K22), Enterobacter
hormaechei SS150, Staphylococcus sciuri SS120, and two Bacillus cereus (SS18 and SS17), demonstrated PGPB-related activities. In addition, they tested the Pseudomonas ae
ruginosa SS183 strain, which produced the ACC deaminase enzyme, being highly antagonistic against Fusarium sp. Authors discussed the synergistic microbial activities and the potential use of this microbial consortia as bioinoculant to be apply as biofertilizer and for the biocontrol of Fusarium sp., under phytopathogen management programs. For this, the manuscript should be accepted for publication, after revision.
During the review, I highlighted the points to be reviewed.
The title does not reflect the objectives, analysis and results of this study. Authors did not perform any biotechnological evaluation, just in vitro assays.
A different title such as
“Potential of tropical bacteria consortia as biofertilizer and Fusarium biocontrol”, will better describe this study
Abstract
Describe the tested treatments and the results probability value, to showed significant differences among activities and production among selected isolates.
Line 35, Change “with” by “showing” in the abstract section.
Rephrase the last paragraph of the abstract, since the statement is rather ambiguous
Introduction
Line 78. Delete “Therefore”
Lines 81-83. How the information of microorganisms establishing relationships with plants is related to the study? Authors tested the selected isolates consortia on plants? Consider deleting this paragraph.
Lines 107-109 This paragraph of the introduction section describes “investing in exploring tropical microbiota transcends the scope of scientific opportunity; it is consolidated as a pressing need to advance toward truly resilient and innovative tropical agriculture” but is not clear how this information is related to this study methodology.
Any information on regard to the phytopathogen fungus selected, Fusarium, was described in the introduction section.
Material and methods
Edit the M&M section to establish the Fusarium source and to explain its culture on PDA was selected to test the bacterial antagonism.
Edit “days” by “d” and “seconds” by “sec” in the M&M section
Discussion
Authors must include information on regard to Fusarium resistance development to chemical fungicides as well, since this was an important part of this study analysis.
I have no other comments on regard to results and conclusion sections.
This manuscript duplicates the numbers in the references section
Reviewer 2 Report
Comments and Suggestions for Authors Dear Editor and AuthorsThe paper titled "Biotechnological potential of tropical bacteria for plant growth promotion and Fusarium suppression" appears to be moderately interesting for readers of Microorganisms, primarily due to its focus on the issue of unavailable phosphorus forms in tropical climates. However, in my opinion, the study is largely preliminary/screening in nature, lacking attempts to verify the functionality of the isolated PGPB - for instance, through phytotron experiments with plants or controlled Fusarium infection. Such experiments could have explored not only the isolates' effects on plant growth or Fusarium suppression but also elucidated the underlying mechanisms, such as whether PGPB application increases the expression of genes related to, e.g., induced systemic resistance (ISR) in plants. Additionally, in terms of biocontrol, the paper even does not clarify whether the isolates produce antibiotic substances (e.g., lipopeptides or polyketides), chitinases, or volatile organic compounds (VOCs). Based on the above, in my opinion, the paper is not suitable for publication in Microorganisms.
Round 2
Reviewer 2 Report
Comments and Suggestions for Authors
Thank you for the explanations, and I accept this article in its current form